# S-Nitrosocysteine Modulates Nitrate-Mediated Redox Balance and Lipase Enzyme Activities in Food-Waste-Degrading *Burkholderia vietnamiensis* TVV75 to Deter Salt Stress

**DOI:** 10.3390/microorganisms13112559

**Published:** 2025-11-10

**Authors:** Youn-Ji Woo, Da-Sol Lee, Ashim Kumar Das, Geum-Jin Lee, Bong-Gyu Mun, Byung-Wook Yun

**Affiliations:** 1Department of Applied Biosciences, College of Agriculture and Life Sciences, Kyungpook National University, Daegu 41566, Republic of Korea; yjwoo2363@gmail.com (Y.-J.W.); giftanna@naver.com (D.-S.L.); ashim@knu.ac.kr (A.K.D.); gj8813@naver.com (G.-J.L.); 2Department of Environment and Biological Chemistry, Chungbuk National University, Cheongju 28644, Republic of Korea; munbg@chungbuk.ac.kr

**Keywords:** nitric oxide, ROS detoxification, nitrogen metabolism, salt stress, lipase activity, microbes

## Abstract

Nitric oxide (NO), a reactive nitrogen species (RNS), plays a role in multiple biological functions and signal transduction. However, the mechanisms by which NO counteracts stress tolerance in microbes have been poorly explored. In addition, the decomposition of salty food waste poses a significant challenge for food-degrading microbes. Therefore, we investigated how S-nitrosocysteine (CysNO) affects the cellular salt stress response of *Burkholderia vietnamiensis* TVV75, a strain isolated from a commercial food waste composite. Under the additional 2% NaCl treatment, increased reactive oxygen species (ROS) inhibited bacterial cell growth and viability. In contrast, CysNO treatment alleviated the cellular ROS levels and growth inhibition by augmenting the superoxide dismutase (SOD) and catalase (CAT) activities. CysNO supplementation also promotes the nitrate reduction pathway in *B. vietnamiensis* TVV75 under salt stress, suggesting NO-mediated nitrogen metabolism for microbial adaptation to salt stress. Furthermore, CysNO restored the intracellular lipid-degrading lipase enzyme activities, which were compromised by salt stress alone. This restoration was accompanied by a concentration-dependent increase in the relative expression of the *lipA* (*lipase A*) and *ELFPP* (*esterase lipase family protein*) genes. These results suggest that external NO supplementation can regulate redox balance, nitrate reduction, and lipase activity to maintain microbial cell growth in high-salt environments, pinpointing a NO-dependent salt stress adaptation strategy for salt-sensitive microbes involved in food waste decomposition.

## 1. Introduction

Global food demand is expected to increase by approximately 56% by 2050 [1], while currently about 20% of the food produced by humanity is lost or discarded [2]. Methods for treating food waste include microbial decomposition, drying and grinding, and sewage mixing treatment. Among these, microbial-based treatment is evaluated as an alternative in terms of economic efficiency and environmental friendliness [3].

In Korea, technologies utilizing microbial fermentation and degradation metabolic processes are employed to convert food waste into feed, compost, biogas, and other products [4]. These microbial-based technologies are applied not only to industrial composting but also to household food waste processors, with their adoption rapidly expanding [5,6]. Home microbial food waste processors utilize composite microbial preparations—mixtures of complex microbial consortia with organic carriers such as sawdust, pulp, or rice husks—to create an environment conducive to microbial growth, thereby efficiently decomposing and reducing household food waste [7,8].

The salt content of Korean food waste is reported to be 2.19%, which is higher than in other countries [9], and is a negative factor for microbial-based food degradation. Salt stress induces osmotic imbalance, protein denaturation, and loss of membrane potential in microbial cells [10,11]. Membrane instability is due to electron leakage caused by oxidation of the membrane [12,13]. This stress not only causes physical damage to the cell but also induces alterations in protein conformation and function, DNA impairment, and leads to decreased cell viability and enzyme activity [11,14], affecting the production of food-degrading enzymes [9,15].

NO is a typical form of reactive nitrogen species (RNS) and is involved in signal transduction and stress regulation in plants, animals, and microorganisms [16,17,18]. In plants, its multifunctional roles have been documented, including the acquisition of salt tolerance through the processing of external nitric oxide, modulation of intracellular redox status, reorganization of antioxidant networks, and regulation of enzymes through protein S-nitrosylation [19,20,21,22]. In particular, NO is a key intermediate in the microbial nitrogen cycle, and microbial growth is tightly controlled in a NO concentration-dependent manner [23]. At low concentrations, NO acts as a signaling molecule to regulate pathogenicity and induce biofilm formation and dispersal [24,25], and at high concentrations, antimicrobial effects through cytotoxicity have been confirmed [26,27]. Moreover, earlier studies reported that NO application enhances the anoxic and heat tolerance in *Escherichia coli* and *Pleurotus ostreatus*, respectively [28,29]. In addition, some studies have reported that exogenous NO induces a reduction in oxidative stress and promotes growth recovery in *E. coli* [30]. It has been reported that NO increases oxidative stress tolerance and pathogenicity through the regulation of *soxR* in *E. coli* [31]. Gusarov et al. [32] also showed that NO can reduce oxidative damage in microorganisms by inhibiting the Fenton reaction. In addition to ameliorating the stresses, NO has also been shown to increase several enzyme activities. Endogenous NO increases the cellulolytic enzyme activity of fungi *Neurospora crassa* [33], and Mesut Taskin et al. found that *Bacillus subtilis*, *Geobacillus stearothermophilus*, and *Yarrowia lipolytica* increase lipase production by exogenous NO supply through sodium nitroprussides (SNPs) [34]. Specifically, lipase is a key enzyme that breaks down lipids in food.

Although NO has been reported to enhance microbial tolerance and stimulate hydrolytic enzyme activities, the stress-reducing effects of NO in microbes under salt stress remain poorly understood. Understanding how NO regulates the physiological and metabolic adaptations of microbes under salt stress is important as a new research axis in microbial-based food degradation. In this study, we focused on analyzing the molecular mechanisms by which the application of exogenous S-nitrosocysteine (CysNO) can improve the salt tolerance of food-degrading bacteria. CysNO has been widely used in mammalian cells and plant studies, demonstrating its broad applicability as a biologically relevant NO donor with non-toxic characteristics compared to SNP. We isolated a single strain through screening from a commercial food waste decomposer preparation (HURIEN, Korea) used in household microbial food processors. Moreover, we performed a series of experiments to assess bacterial growth, cell viability, ROS production, nitrogen metabolism, and lipase activity under salt-stressed conditions with CysNO supplementation.

## 2. Materials and Methods

### 2.1. Isolation and Identification of Food Waste-Degrading Bacteria

The HURIEN food waste decomposer, specifically Model HD-N100PD, is a home appliance designed for processing kitchen waste, primarily using a microbial decomposition product. Bacteria were isolated from this commercial food waste decomposing product by suspending 5 g of the sample in 200 mL of sterile distilled water and incubating at 37 °C, 150 rpm for 24 h. After pre-enrichment, 100 μL of culture was spread on tryptic soy agar (TSA; Difco, Franklin Lakes, NJ, USA) and incubated at 30 °C for 24–48 h. A dominant colony was selected and purified for further analysis. For identification, the 16S rRNA gene was amplified, and PCR products were purified, sequenced, and compared using (National Center for Biotechnology Information) NCBI (Basic Local Alignment Search Tool) BLAST (https://blast.ncbi.nlm.nih.gov/Blast.cgi, accessed on 29 September 2025). Species-level identity was assigned based on ≥99% sequence similarity. A maximum likelihood phylogenetic tree of the *B. vietnamiensis* TVV75 together with the associated sequence was created using Unipro UGENE v52.1 with 1000 bootstrap replications [35].

### 2.2. NO-Treated Bacterial Growth Assessment Under Salt Stress

To assess the response of NO-mediated salt stress, the isolate was cultured in two conditions. The control was cultured in tryptic soy broth (TSB) (Difco, USA) with 0.5% NaCl, while bacteria under salt stress were cultured in TSB with 2% NaCl. This group simultaneously supplemented with 0, 0.05, 0.1, and 0.25 mM of CysNO. The additional 2% NaCl concentration was selected based on a preliminary test (see Appendix A). TSB was cultured at 30 °C, 150 rpm for 30 h. OD_600_ value was measured every 6 h. Tryptic soy agar (TSA) was incubated at 30 °C for 6 days, and colony diameters were measured using a digital caliper. Each condition was tested in triplicate, and the mean OD value and colony size were used to compare growth.

### 2.3. CYTO9/PI Cell Viability Assay

Bacterial viability was determined using the SYTO 9/propidium iodide (PI) Live/Dead BacLight kit (Invitrogen, L7012, Carlsbad, CA, USA) [36]. Cells were cultured for 24 h and washed twice with 0.9% NaCl, resuspended, and stained with SYTO 9/PI mixture (3 µL/mL) for 15 min in the dark. Fluorescence images were captured by Nikon ECLIPSE Ti2-U, Tokyo, Japan (SYTO 9: Ex 480/Em 500–540 nm; PI: Ex 535/Em 600–650 nm). Quantification of the fluorescent area was performed using ImageJ (Version 1.54r), and viability (%) was calculated as the ratio of SYTO 9–positive/PI–negative fluorescence to total stained cells. Ethanol-treated cells served as dead controls, and fresh cultures as live controls.

### 2.4. Enzyme Extraction

Enzyme extraction was performed following the methodology of Doerner et al. [37]. Briefly, after 30 h of bacterial incubation, cultures were centrifuged at 3000 rpm for 20 min at 4 °C, and the supernatant was collected as the extracellular enzyme fraction. The pellet was washed with PBS and resuspended in lysis buffer containing 50 mM Tris-HCl (pH 7.5), 0.5% (*w*/*v*) Triton X-100, and 0.05% (*w*/*v*) lysozyme, followed by incubation at 37 °C for 30 min. Lysates were centrifuged at 13,000 rpm for 10 min at 4 °C to obtain the intracellular enzyme fraction. Protein content was quantified using the Bradford assay with bovine serum albumin (BSA, Thermo Scientific, Waltham, MA, USA) as the standard. All procedures were performed in triplicate.

### 2.5. Nitrate Reductase Activity and Nitrite Determination

Nitrate reductase (NR, EC 1.6.6.1) activity was determined by quantifying nitrite production using the Griess reaction [38]. Briefly, 200 μL enzyme extracts were incubated with assay buffer (100 mM potassium phosphate, pH 7.6, 40 mM NaNO_3_, 0.2 mM NADH) at 24 °C for 15 min. The reaction was stopped with 100 μL 0.5 M zinc acetate, followed by the addition of 400 μL Griess reagent (modified; Sigma-Aldrich, G4410, St. Louis, MO, USA). After 15 min in the dark, samples were centrifuged at 14,000 rpm for 2 min, and absorbance was measured at 540 nm. Nitrite content was assayed by the Griess reaction [39]. A total of 100 μL of cell culture was incubated with 100 μL Griess reagent, and absorbance was measured at 540 nm. Sodium nitrite was used for the standard curve.

### 2.6. Measurement of Hydrogen Peroxide and Malondialdehyde

Hydrogen peroxide (H_2_O_2_) content was determined according to Velikova [40], with modifications. Cellular precipitate was homogenized in 0.6 mL of 0.1% (*w*/*v*) trichloroacetic acid (TCA) and centrifuged at 11,500 rpm for 15 min. For the assay, 80 μL of supernatant was mixed with 200 μL of 100 mM potassium phosphate buffer (pH 7.5) and 800 μL of 1 M potassium iodide (KI). The mixture was incubated on ice for 1 h, followed by 20 min at room temperature. Absorbance was measured at 390 nm. Moreover, malondialdehyde (MDA) content was measured following the method of Heath and Packer [40], with slight modifications. The same extract was used, and 400 μL of supernatant was mixed with 1 mL of 20% TCA containing 0.5% thiobarbituric acid (TBA). The mixture was heated at 95 °C for 30 min and cooled on ice. Absorbance was read at 532 nm and 600 nm, and MDA concentration was calculated using an extinction coefficient of 156 mM^−1^ cm^−1^.

### 2.7. Antioxidant Activity Assay

Catalase (CAT, EC 1.11.1.6) activity was determined by monitoring the decomposition of H_2_O_2_ using OxiTec^TM^ Catalase Assay Kit (Colorimetric/Fluorometric; BIOMAX Co., Ltd., Seoul, Republic of Korea; cat. no. BO-CAT-400). Enzyme extracts (25 µL) were mixed with 25 µL of 40 µM H_2_O_2_ in a 96-well plate and incubated for 30 min at room temperature in the dark. After the reaction, 50 µL of a working solution containing 0.5 µL of Oxi-probe, 0.2 µL of horseradish peroxidase (100 U/mL), and 49.3 µL of 1× reaction buffer was added. The mixture was incubated at 37 °C for 30 min in the dark, and absorbance was measured at 570 nm.

Superoxide dismutase (SOD, EC 1.15.1.1) activity was assessed by measuring the inhibition of NBT photoreduction [41]. A reaction mixture containing 50 μL of enzyme extract, 13 mM methionine, 75 μM nitroblue tetrazolium (NBT), 2 μM riboflavin, and 0.1 mM EDTA in 50 mM potassium phosphate buffer (pH 7.8) to a final volume of 1 mL was prepared. The mixture was incubated at 25 °C under 4000 lux light for 15 min. Absorbance was read at 560 nm using 200 μL of the reaction mixture. A sample without light or enzymes served as the blank; 50 mM potassium phosphate buffer (pH 7.8) was used as the positive control. One unit (U) of SOD was defined as the amount causing 50% inhibition of NBT reduction. Activity was expressed as U mg^−1^ protein, normalized to the protein content determined by the Bradford assay.

### 2.8. Lipase Activity Assays

Lipase was measured using p-nitrophenyl palmitate (pNPP) as a substrate [42]. A mixture of 30 mg pNPP (in 10 mL isopropanol) and 90 mL of 50 mM potassium phosphate (pH 9.0) with 200 µL Triton X-100 was used. Enzyme extract (100 µL) of cell culture supernatant (secreted extracellular enzymes) was added to 2.4 mL substrate and incubated at 37 °C for 30 min. Absorbance was read at 410 nm (ε = 2.71 × 10^3^ M^−1^ cm^−1^). For clear zone evaluation, 1% tributyrin in nutrient agar plates were used [43]. A single colony was grown at 30 °C for 6 days. Activity index (AI) was calculated as the ratio of the clear-zone diameter to colony diameter.

### 2.9. Relative Gene Expression Analysis

Total RNA was extracted using the AccuPrep^®^ Universal RNA Extraction Kit (Bioneer, Daejeon, Republic of Korea) according to the manufacturer’s instructions. Complementary DNA (cDNA) was synthesized from 1 μg of total RNA using the Solg^TM^ RT kit (Solgent, Daejeon, Republic of Korea). Quantitative real-time PCR was conducted using the CFX Duet Real-Time PCR System (Bio-Rad, Hercules, CA, USA) with the Solg^TM^ 2× Multiplex Real-Time PCR Smart mix (Solgent, Daejeon, Republic of Korea) containing SYBR Green. All tests were performed in triplicate. Relative gene expression levels were determined based on threshold cycle (Ct) values. Primers are listed in Appendix B.

## 3. Results

### 3.1. Isolation and Molecular Identification of Strains

To choose a bacterial strain for salt stress analysis, bacterial strains were isolated from a commercial microbial food waste-degrading product using TSA medium. The dominant colony was selected and identified by 16S rRNA gene sequencing, which showed 100% identity with *Burkholderia vietnamiensis* strain TVV75 in the NCBI type strain 16S rRNA database using the BLASTN 2.13.0+ program. Phylogenetic analysis confirmed that the isolate belonged to the *B. vietnamiensis* clade (Figure 1). *B. vietnamiensis* has previously been reported for its tolerance to various environmental stresses and can degrade lipases [44,45,46]; therefore, this strain was selected as a biological model for further investigation of stress responses and lipase activity under saline conditions and supplemented with different concentrations of CysNO.

### 3.2. CysNO Protected B. vietnamiensis TVV75’s Growth Under NaCl Stress

Food-waste-degrading microbes are often challenged by salt stress that prevents a smooth decomposition process [9]. Nonetheless, studies have also found food-waste-degrading bacteria that tolerate salt stress [47]. In this study, we aimed to test whether our isolated strain *B. vietnamiensis* TVV75 survives under salt stress conditions. After 30 h of incubation, we noticed notable growth inhibition of *B. vietnamiensis* TVV75 under TSB + 2% NaCl, indicating an intolerant phenotype of our selected strain (Figure 2A,B). Along with or without salt stress, we applied 0, 0.05, 0.1, and 0.25 mM of CysNO and cultured the bacteria for 30 h. The addition of 0.05 and 0.1 mM of CysNO showed improved OD_600_ levels under both conditions, which pinpoints that CysNO potentially helps *B. vietnamiensis* TVV75 to sustain its cell growth. However, a relatively high CysNO concentration (0.25 mM) displayed an inhibitory result in bacterial growth (Figure 2A,B). Overall, these findings indicated that a lower concentration of CysNO can mitigate the inhibitory effect of NaCl on the food-waste-degrading strain, *B. vietnamiensis* TVV75.

### 3.3. CysNO Reduced Cell Death Under NaCl Stress

Given the observed growth inhibition under salt stress, we tested whether this was accompanied by reduced cell viability rather than just a growth reduction. We performed SYTO 9/propidium iodide (PI) live/dead staining and imaged with fluorescence microscopy. Viable cells emitted green fluorescence (SYTO9), whereas non-viable cells emitted red fluorescence (PI) [48]. Merged channel images were used to compare viability with or without salt stress supplemented with a series of CysNO concentrations. *B. vietnamiensis* TVV75 cultured in additional 2% NaCl showed a predominance of PI and less SYTO9 fluorescence compared with the absence of salt stress (Figure 3A). In contrast, CysNO-treatments largely reduced the PI fluorescence in both the presence and absence of salt stress conditions, indicating that NO mitigates the salt-induced negative impacts on cell growth and viability.

Quantitative analysis of the SYTO9/PI fluorescence area ratio of these fluorescence signals supported the results of the microscopy images. TSB alone, with no CysNO-treated groups, had approximately 30% of PI-positive cells, indicating the presence of underlying membrane damage (Figure 3B). As the CysNO treatment concentration increased from 0.05 to 0.25 mM, the percentage of PI-positive cells decreased, falling below 10% at 0.05 mM, and remained low until 0.25 mM (Figure 3B). On the other hand, under additional 2% NaCl salt stress conditions, the PI-positive cells increased significantly to about 40%, indicating a significant increase in salt stress-induced cell death. However, the PI-positive percentage gradually decreased with the increase in CysNO concentration, indicating that salt-induced cell death was notably alleviated by NO (Figure 3B).

### 3.4. CysNO Application Improved ROS Detoxification Under NaCl Stress

ROS are widely used as indicators of cellular stress, particularly for microbes, where excessive ROS is lethal to their community [49]. NO, a mediator of redox switch in the cell, balances the ROS accumulation in an antioxidant-dependent manner [50]. Thus, we analyzed oxidative stress induced by salt stress and its alleviation by CysNO in *B. vietnamiensis* TVV75. Under salt stress, H_2_O_2_ and MDA increased in the culture of *B. vietnamiensis* TVV75 more than 4-fold relative to the control. In contrast, H_2_O_2_ and MDA levels were largely decreased in *B. vietnamiensis* TVV75 under salt stress by the supplementation of CysNO, except for H_2_O_2_ in 0.05 mM of CysNO (Figure 4A,B). To investigate whether CysNO mitigates ROS through antioxidant activity, we measured the activity of SOD and CAT. We also measured transcriptional changes in the genes of *osmC1*/*C2* (*osmotically inducible protein C*) and *treZ* (*maltooligosyltrehalose trehalohydrolase*), which are the proteins that help bacterial cells respond to internal pressure and damage under oxidative and osmotic stress, respectively. We found an increased activity of SOD and CAT with the supplementation of CysNO, particularly at 0.05 and 0.1 mM of CysNO (Figure 4C,D). However, at a relatively higher concentration (0.25 mM), their activities were decreased, which seems to contradict earlier findings on lower ROS production. The reduced ROS production may be directed by the upregulation of *treZ* at 0.25 mM of CysNO rather than the SOD and CAT activity (Figure 4G. Nevertheless, consistent with SOD and CAT activity, the relative expression of *osmC1*/*osmC2* showed a significant upregulation by 0.1 mM of CysNO treatment (Figure 4E,F). Collectively, these results indicated that NO balances the redox switch in *B. vietnamiensis* TVV75 under salt stress, which mitigates cellular damage and parallels a reduction in cell death.

### 3.5. CysNO Application Altered Nitrogen Metabolism to Protect the Bacterial Cells Under NaCl Stress

Saline conditions heavily interrupt bacterial nitrogen metabolism, a crucial factor in maintaining a sustainable ecosystem [47,51]. In particular, nitrate reduction plays a major role in the nitrogen cycle, whereby nitrate is sequentially reduced to nitrite, N oxides (NO and N_2_O), and dinitrogen (N_2_) [52]. Therefore, to assess whether CysNO changes in nitrate reduction pathways (respiratory vs. assimilatory) under salt stress, we quantified NR activity, extracellular NO_2_^−^, and transcriptional changes in *narG*, *narH* (membrane nitrate reductase), *hmp* (flavohemoglobin for NO detoxification), and *nirB* and *nirD* (assimilatory nitrite reductase) in *B. vietnamiensis* TVV75 (Figure 5). Results exhibited that CysNO treatments had a noteworthy modification in the reduction process with or without stress conditions. A NO reducer to nitrate, *Hmp*, was notably upregulated with the series of CysNO treatment under salt stress (Figure 5A). Subsequently, we noticed higher NR activity and relative expression of *NarG* and *NarH* in the bacterial culture when supplemented with CysNO concentrations, highlighting a smooth conversion of nitrate to nitrite by exogenous NO application (Figure 5B–E). Consistently, nitrite reductase genes (NirB/D) were induced by CysNO in a concentration-dependent manner, followed by a slight decline at the highest concentration (Figure 5D,F). Together, these results support that CysNO application balances the nitrogen metabolism in *B. vietnamiensis* TVV75 under salt stress.

### 3.6. CysNO Application Improved Lipase Activity Under NaCl Stress

With the observation of CysNO-induced NaCl tolerance in *B. vietnamiensis* TVV75, lipase activity was quantified under salt-stress conditions to verify whether CysNO supplementation could restore salt-inhibited lipid degradation and enhance food waste degradation efficiency. We quantified lipase activity with p-nitrophenyl palmitate (pNPP) as a substrate, measured lipase-related gene expression, and performed tributyrin agar assays. Intracellular activities were determined from cell lysates, and culture supernatants were analyzed for extracellular activities. Under salt stress, intracellular lipase activity increased 4-fold relative to the control condition, where CysNO further enhanced these intracellular activities in a concentration-dependent manner across the tested range (Figure 6A). Consistently, under salt-stress conditions, the relative expression of *lipA* (triacylglycerol lipase A) and *ELFPP* (esterase lipase family protein) increased with the elevation of CysNO concentrations (Figure 6B,C). In contrast, we did not notice any notable difference in extracellular lipase activity with or without salt stress or by CysNO application, except for 0.1 mM CysNO. Furthermore, on tributyrin agar, colony diameter and the absolute size of the clear zone were decreased under salt stress, whereas the Activity Index (AI; clear zone–colony size ratio) was increased (Figure 6E–H), suggesting salt-induced suppression of cell growth that lowered cell density in the culture and thereby reduced the total amount of secreted extracellular lipase. Except for colony diameter, the clear zone and AI were found to be increased by CysNO application, indicating a positive regulation of lipase activity by NO.

## 4. Discussion

Food is a basic human need, yet we face a significant challenge with food waste from households, markets, and restaurants [53,54]. This problem is exacerbated by the high intrinsic water and salinity content in food waste, which hinders smooth decomposition [9,55]. Except for a few salt-tolerant microbes, excess salinity induces lysis in salt-sensitive microbes, resulting in reduced cell integrity [56]. In this study, we isolated *B. vietnamiensis* TVV75 from a microbial food waste composting product and noticed its growth and viability were notably disrupted under 2% NaCl added stress conditions (Figure 2). These results are in alignment with the finding that salinity is a critical factor for food-degrading microbes. However, the salt-treated bacterial culture was supplemented with a series of CysNO concentrations to determine whether NO outperforms the negative impact of salt stress. NO, a small redox molecule with multivariant functions, commonly protects against oxidative damage, regulating signaling transduction and growth and development under various abiotic and biotic stressors [57,58,59,60].

We observed that 0.05 and 0.1 mM of CysNO restored salt-inhibited bacterial growth constantly until 30 h (Figure 1). The results provide a preliminary understanding that NO can mitigate the growth reduction of *B. vietnamiensis* TVV75 caused by salt stress. Then, we checked the bacterial cell viability using a SYTO9/PI fluorescence-based cell viability assay. Surprisingly, the bacterial cultures treated with CysNO, both with and without salt, showed a strong fluorescence of SYTO9, which indicates a high number of living cells. In contrast, a lower PI fluorescence suggests that NO positively reduced salt-induced cell death (Figure 3). In the absence of CysNO, living cells were largely absent under NaCl, pinpointing that salt stress hinders bacterial survivability. Consistently, salt treatment significantly increased H_2_O_2_ and MDA levels, suggesting ROS production that impaired membrane integrity (Figure 4A,B). High salt perturbs microbial water activity and ionic homeostasis, causing fluctuations in membrane fluidity and potential, and induces ROS accumulation through reduced efficiency of electron transport chains and metal-catalyzed reactions [61]. In addition, under salt conditions, the disruption of iron homeostasis and the accumulation of H_2_O_2_ overlap and may promote the Fenton reaction (Fe^2+^ + H_2_O_2_ → -OH + OH^−^ + Fe^3+^) in bacteria, where the generated -OH is highly reactive and leaves irreversible damage to membrane lipids and protein-nucleic acids [61,62]. The increased H_2_O_2_ and MDA, along with increased dead cells observed in our experiments, can be interpreted as a result of this Fenton-weighted oxidative load and disruptions of the bacterial respiratory electron transport chains. On the other hand, CysNO significantly lowered H_2_O_2_ and MDA levels. These changes induced by NO treatment can be explained by the previous studies [32,39]. NO inhibits the accumulation of bacterial superoxide and lipid peroxidation by blocking chain radical reactions and reorganizing the antioxidant network, and NO temporarily inhibits the reductive flow of free thiol centers, limiting Fe^3+^ reductions to Fe^2+^ and accelerating H_2_O_2_ removal through maintenance/reactivation of catalase activity, resulting in the simultaneous depletion of Fe^2+^ and H_2_O_2_, the fuel for the Fenton reaction. Together, these two actions slow down -OH production and mitigate membrane, protein, and nucleic acid damage; therefore, the concomitant decrease in PI-positive ratio, H_2_O_2_, and MDA observed in this study corresponds to this.

*B. vietnamiensis* TVV75 also affected the nitrate reduction pathway under salt stress, which was turned over by the supplementation of CysNO. CysNO application upregulated the transcriptional levels of *hmp*, *narG*, and *narH*, together with NR activity, suggesting rapid NO reduction to NO_2_- via NO_3_-. A parallel elevation of NR activity and *narH* transcription levels has been reported in *Halomonas* sp. B01 under salt stress [63]. Increased NR activity does not lead to NH_4_^+^ accumulation, resulting in continuation of the nitrate–nitrite–NO cycle (Figure 5F,G). This is consistent with the observation that nirB/D gene expression remained unchanged under salt stress. It may be possible that lowering the Fenton-driven oxidative load by NO alleviates the vicious circle of Fe-S protein damage and OxyR/SoxRS hyperactivity, allowing the nitrate respiration and Hmp-mediated NO detoxification pathways to maintain their function, stabilizing the nitrogen species flow [32]. Furthermore, reports are showing that the presence of nitrate or nitrite under anoxic conditions strongly induces *hmp* [64]. This NR- and *hmp*-induced nitrate–NO cycle efficiently alleviates stress by replenishing the cell’s energy deficit while maintaining redox balance through RNS regulation. However, in our study, treatment with 0.25 mM of CysNO saturated the gains in NR and *hmp* transcripts (Figure 5A,B), suggesting that excessive NO may have increased the nitrosylation/nitrosation burden [65] or exceeded the capacity of *hmp* processing, thereby offsetting the net effect [66]. This disruption of redox balance may induce RNS toxicity and impair cellular activity, which may explain the lack of salt stress reduction.

These molecular responses strongly support the involvement of NO signaling through CysNO. However, the potential bioactivity of the donor, cysteine, cannot be entirely excluded. Previous studies have reported that cysteine can influence microbial growth and metabolism, but such effects were generally observed at high concentrations (0.4–10 mM) [67,68,69]. In contrast, the highest concentration of CysNO used in this study (0.25 mM) was far below this range, and the growth-promoting effect even declined at higher doses. This pattern suggests that the physiological recovery observed here was not driven by cysteine supplementation but rather by the regulation of NO-mediated signaling.

To further clarify this distinction, additional experiments comparing cysteine alone with other NO donors—such as sodium nitroprusside (SNP) and S-nitrosoglutathione (GSNO)—would be valuable. Because physiological responses to NO can differ depending on donor type and chemical stability, such comparative analyses could verify both the generality and mechanistic specificity of NO-dependent regulation. In this context, the present study proposes an initial framework for understanding the NO-mediated stress alleviation mechanism.

This stabilization at the cellular level is manifested in the recovery of functional output–lipid hydrolysis performance as shown in Figure 6. Intracellular lipase activity increased 4-fold under salt conditions and was further elevated by CysNO treatment, and *lipA* and *ELFPP* transcripts increased at 0.05 mM and 0.1 mM CysNO. However, at 0.25 mM CysNO treatment, *lipA* transcription levels decreased under salt stress, but total intracellular lipase activity increased. This discrepancy may reflect the inherent differences between mRNA and enzymes. The half-life of bacterial mRNA is extremely short, whereas the enzyme is relatively stable. In the study of *Bacillus* sp. LBN 2 cultures, while lipase activity peaked at 48 h, a substantial amount of enzyme protein persisted until 72 h [70]. Therefore, the total activity of pre-accumulated enzymes can be measured at a high level even if *lipA* transcription is temporarily low at a specific point in time. Moreover, the *B. vietnamiensis* gene encodes multiple lipolytic enzymes, including *ELFPP*, which can likely contribute to total lipase activity rather than a single *lipA* product. On tributyrin agar, salt decreased colony size and absolute clear zone but increased Activity Index (AI), which was further enhanced by CysNO. This suggests that salt stress may reduce the absolute amount of secreted enzyme through growth inhibition, but CysNO restored membrane stability and viability and optimized nitrogen/NO metabolism, resulting in higher efficiency per unit biomass. Similarly to our results, Taskin et al. [34] reported that SNP treatment enhanced the lipase activity.

Our present study was conducted on a single strain of *Burkholderia vietnamiensis* TVV75, which has been established as a model system to characterize NO-specific responses from both quantitative and molecular perspectives. The results suggest the molecular mechanisms by which NO affects the physiological adaptation of microorganisms under salt stress induce their lipase activity. However, in real food degradation, not only lipases but also many enzymes are involved, including amylases, proteases, and cellulases [71,72], and the effect of NO on these hydrolytic enzymes is lacking. Furthermore, food composting processes occur in complex microbial consortia where different bacteria and fungi coexist and interact. In these ecosystems, the metabolic networks of individual species work complementarily. These dynamic interactions can significantly impact overall degradation efficiency, oxidation reduction balance, and stress tolerance [73,74]. Therefore, it is necessary to validate the ecological validity and generality of the NO-dependent regulatory mechanisms proposed in this study through mock experiments involving multiple strains from the same commercial preparation or based on mixed culture systems.

Furthermore, the strain used in our study, *B. vietnamiensis*, belongs to the *Burkholderia cepacia* complex (BCC), a group comprising both environmental and opportunistic clinical species. Within this complex, *B. vietnamiensis* is considered to exhibit relatively low pathogenic prevalence, as infections in cystic fibrosis patients have been reported only occasionally [75]. Thus, caution is advised for potential use in large-scale or commercial applications. Aside from this biosafety consideration, this study is limited, as only an in vitro-level experiment was conducted under controlled culture conditions, which does not fully reflect complex environmental variables such as temperature, oxygen concentration, substrate composition, and microbial interactions. Future studies should validate the applicability of the mechanisms proposed in this study by evaluating the effects of NO treatment in consortium-based pilot-scale systems that mimic the environment of a real household digester. In addition, to extend the understanding of the mechanisms of NO in salt-sensitive, food-waste-degrading microorganisms and to use them in practical food waste processing, hmp or NR-deleted strains or the application of NO scavenging product will be a possible approach. Such future studies will further support the potential application of nitric oxide-mediated enhancement of microbial salt tolerance in real industrial conditions.

## 5. Conclusions

Our study suggested that exogenous CysNO alleviates the oxidative load in *B. vietnamiensis* TVV75 induced by salt stress. Moreover, CysNO induced nitrate reductase activity and *hmp* expression stabilized the nitrogen cycle, restoring cell survival, membrane integrity, and lipid metabolism. However, the effect of CysNO was concentration-dependent; low concentrations of CysNO induced protection, while high concentrations reversed it. Despite the effective salt tolerance by NO supplementation in a single strain, our findings left open the question of how NO supplementation reacts across multiple microbial strains with different salt, pH, and temperature conditions. Future validation under full-scale composting with microbial consortia or household processor conditions will be crucial to assess the practical relevance of CysNO-mediated composting enhancement. The effect of NO should also be checked not only for lipase but also for amylase, protease, and cellulase in food-degrading microbes. Collectively, this present study suggested that a lower concentration of CysNO protected the cells from salt stress.

## Figures and Tables

**Figure 1 microorganisms-13-02559-f001:**
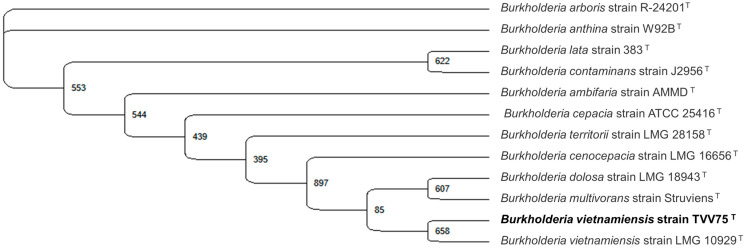
Identification of the food waste microbe isolate as *Burkholderia vietnamiensis* TVV75 by 16S rRNA gene sequence. The maximum likelihood method was employed in Unipro UGENE v52.1 with 1000 bootstrap replications, as shown in the corresponding nodes. *Burkholderia vietnamiensis* TVV75 formed a clade with a common evolutionary relationship.

**Figure 2 microorganisms-13-02559-f002:**
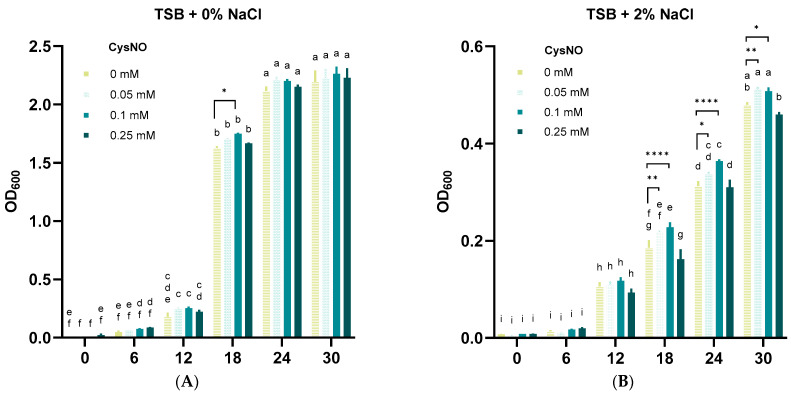
Effects of CysNO on *B. vietnamiensis* TVV75 growth with or without salt stress conditions. (**A**,**B**) Growth rate of *B. vietnamiensis* TVV75 at every 6 h interval in TSB (0.5% NaCl; control) or TSB + 2% NaCl supplemented with 0, 0.05, 0.1, and 0.25 mM of CysNO. Data are shown as mean ± SE (*n* = 4). Different letters denote statistically differences among treatments, performed by two-way ANOVA with Tukey test (*p* < 0.05). Student’s *t*-test (* *p* < 0.05, ** *p* < 0.01, and **** *p* < 0.0001) was also performed to compare 0 mM with each CysNO concentration. Three independent experiments were carried out with similar results.

**Figure 3 microorganisms-13-02559-f003:**
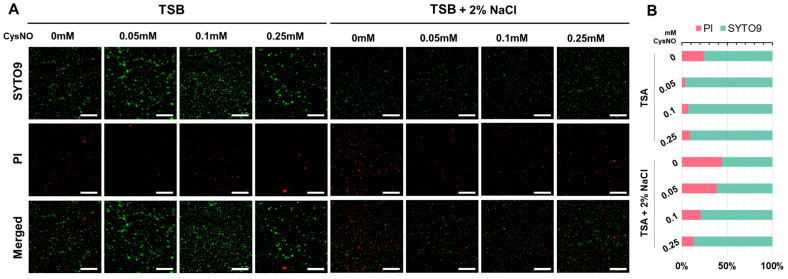
(**A**) SYTO9/PI fluorescence-based (green, SYTO9; red, PI) viability assay of *B. vietnamiensis* TVV75 after 6 days of incubation in TSB and TSB + 2% NaCl supplemented with 0, 0.05, 0.1, and 0.25 mM of CysNO. Scale bars = 100 μm. The representative image was chosen from three individual replicates. (**B**) Quantification under the microscopy image indicates the percentages of PI-positive and SYTO9-positive areas.

**Figure 4 microorganisms-13-02559-f004:**
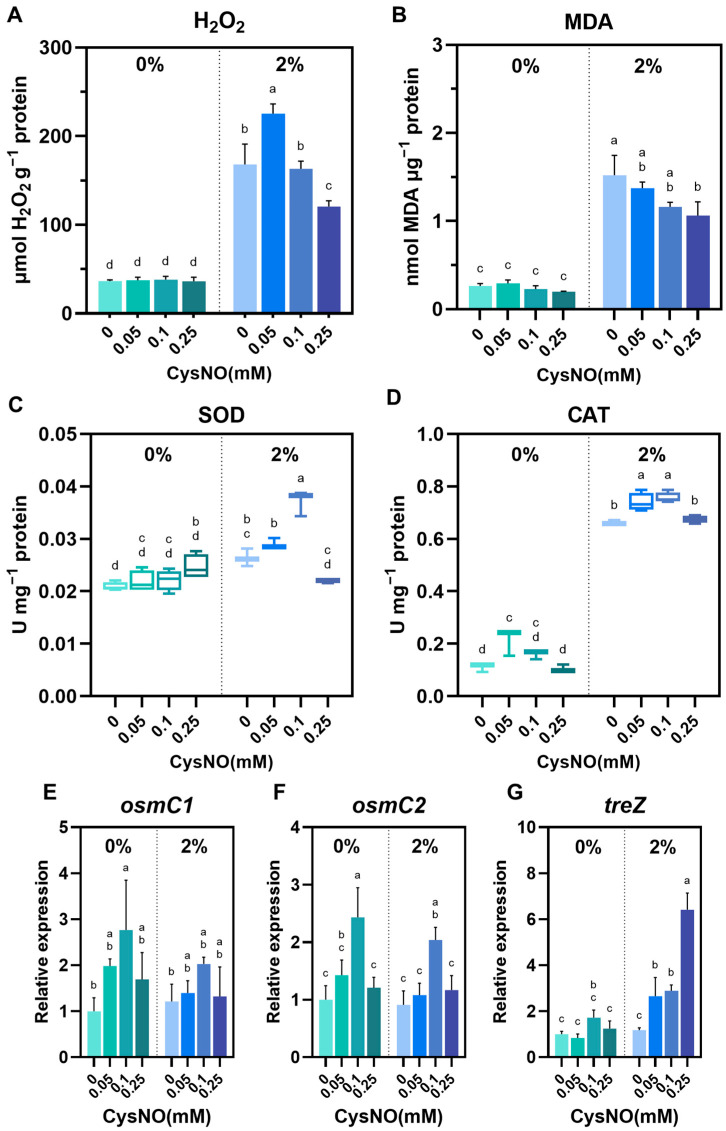
Effect of CySNO on redox homeostasis of *B. vietnamiensis* TVV75 with or without salt stress conditions. (**A**) Intracellular H_2_O_2_ levels, (**B**) malondialdehyde (MDA), (**C**) SOD activity, and (**D**) CAT activity in *B. vietnamiensis* TVV75 after 6 days of incubation in TSB and TSB + 2% NaCl supplemented with 0, 0.05, 0.1, and 0.25 mM of CysNO. (**E**–**G**) Relative expression of *osmC1*, *osmC2*, and *treZ* in *B. vietnamiensis* TVV75 after 6 days of incubation in TSB and TSB + 2% NaCl supplemented with 0, 0.05, 0.1, and 0.25 mM of CysNO. Data are shown as mean ± SE (*n* = 3). Different letters denote statistically differences among treatments, performed by two-way ANOVA with Tukey test (*p* < 0.05). *osmC*, *osmotically inducible protein C*; *treZ*, *maltooligosyltrehalose trehalohydrolase*.

**Figure 5 microorganisms-13-02559-f005:**
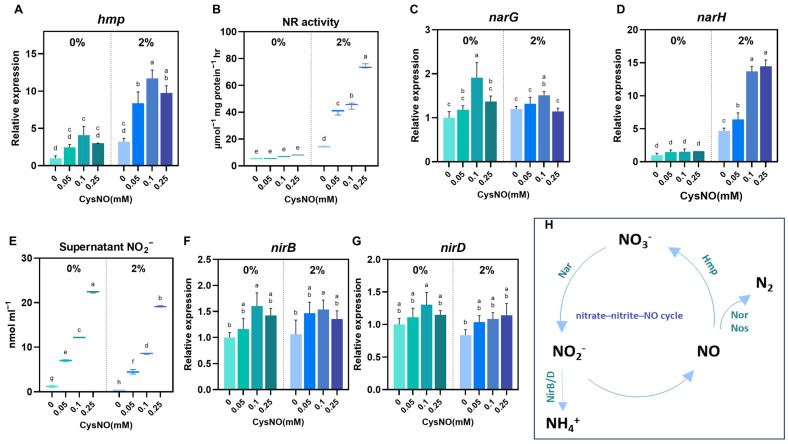
Effect of CySNO on nitrogen metabolism of *B. vietnamiensis* TVV75 with or without salt stress conditions. Relative expression of (**A**) *hmp*, (**C**,**D**) *narG*, *narH*, and (**F**,**G**) *nirB*/*D* in *B. vietnamiensis* TVV75 after 6 days of incubation in TSB and TSB + 2% NaCl supplemented with 0, 0.05, 0.1, and 0.25 mM of CysNO. (**B**) Nitrate reductase (NR) activity and (**E**) supernatant nitrite content in *B. vietnamiensis* TVV75 after 6 days of incubation in TSB and TSB + 2% NaCl supplemented with 0, 0.05, 0.1, and 0.25 mM of CysNO. (**H**) A model of nitrogen metabolism. Data are shown as mean ± SE (*n* = 3). Different letters denote statistically differences among treatments, performed by two-way ANOVA with Tukey test (*p* < 0.05). Nar, membrane nitrate reductase; NirB/D, assimilatory nitrite reductase; Hmp, flavohemoglobin; Nor/Nos, nitric-oxide/nitrous-oxide reductases.

**Figure 6 microorganisms-13-02559-f006:**
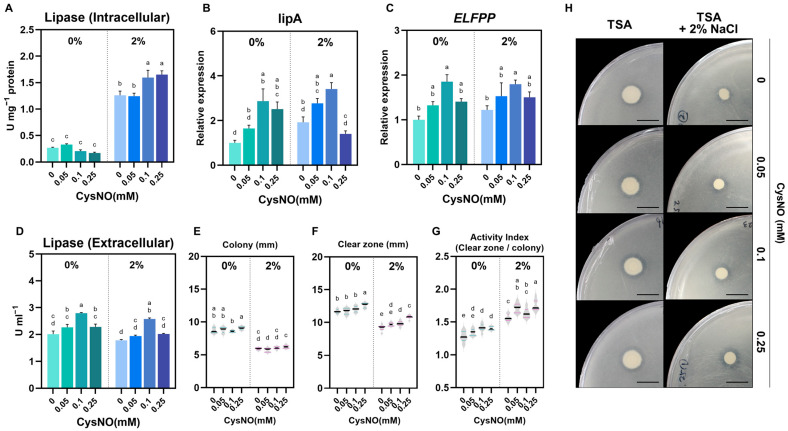
Effect of CySNO on the lipase activity of *B. vietnamiensis* TVV75 with or without salt stress conditions. (**A**) Lipase activity in cell lysate (intercellular) and (**D**) culture supernatants (extracellular) in *B. vietnamiensis* TVV75 after 6 days of incubation in TSB and TSB + 2% NaCl supplemented with 0, 0.05, 0.1, and 0.25 mM of CysNO. (**B**,**C**) Relative expression of *lipA* and *ELFPP* in *B. vietnamiensis* TVV75 after 6 days of incubation in TSB and TSB + 2% NaCl supplemented with 0, 0.05, 0.1, and 0.25 mM of CysNO. (**E**–**G**) Tributyrin agar quantification: colony diameter, clear-zone diameter, Activity Index, and (**H**) representative tributyrin agar images in *B. vietnamiensis* TVV75 after 6 days of incubation in TSB and TSB + 2% NaCl supplemented with 0, 0.05, 0.1, and 0.25 mM of CysNO. Data are shown as mean ± SE (*n* = 3). Different letters denote statistically differences among treatments, performed by two-way ANOVA with Tukey test (*p* < 0.05). lipA, lipaseA; ELFPP, esterase lipase family protein. Scale bars = 1 cm.

## Data Availability

The original contributions presented in this study are included in the article/Appendix A. Further inquiries can be directed to the corresponding author.

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
