# Peer review of "S-Nitrosocysteine Modulates Nitrate-Mediated Redox Balance and Lipase Enzyme Activities in Food-Waste-Degrading *Burkholderia vietnamiensis* TVV75 to Deter Salt Stress"

_microorganisms, 2025, doi:10.3390/microorganisms13112559_

Round 1

Reviewer 1 Report

Comments and Suggestions for Authors

Major points

  1. Include (1) an L-cysteine-only control matched to CysNO, (2) at least one alternative NO donor (e.g., SNP or GSNO), and/or (3) an NO scavenger (e.g., cPTIO). These are needed to separate NO-dependent effects from donor- or cysteine-specific effects.
  2. Unify the analysis across figures: define factors for ANOVA, report assumption checks (normality/homoscedasticity), multiple-comparison procedures, exact n per panel/metric, and whether effects are fixed or random. Ensure consistency between the text and figure legends: t-test figure 2) vs. Tukey (figure 4-6). Why did you use t-test in figure 2?
  3. Resolve inconsistencies in the control medium NaCl content across Methods and figure captions, and clarify whether “+2% NaCl” is total or added NaCl. This affects interpretation of all salt-response data.
  4. Report activity units explicitly (e.g., U·mg⁻¹ protein - fig. 5B), how protein was quantified for each assay, and define clearly what is measured in intracellular vs. extracellular fractions (lysate vs. supernatant). In text we read " One unit (U) of SOD was defined as the amount causing 50% inhibition of NBT reduction. Activity was expressed as U/mL based on a 0.05 mL extract volume", but in figure 4C we can see - U mg protein. Is it correct?

Minor points

  1. Where causality hinges on CysNO without a scavenger/alternate donor, soften to “support/suggest” rather than “demonstrate.”
  2. Unify 37°C or 37 °C (p. 3)
Comments on the Quality of English Language

The manuscript is clearly structured but requires moderate editing for grammar, word choice, and stylistic consistency. A light professional edit by a fluent speaker is recommended.

  1. Use past tense for Methods/Results and present for general facts in Introduction/Discussion.
  2. Replace nonstandard phrases (e.g., “constitutive times” → “independent experiments”).
  3. Keep lists grammatically parallel (e.g., “measured enzyme activity, quantified gene expression, and assessed viability”).
  4. Use non-breaking space between value and unit (e.g., “25 °C,” “2% NaCl” without extra space before “%”); use a period for decimals.

Author Response

Reviewer 1

Thank you very much for your thoughtful and detailed evaluation of our manuscript. Your insightful comments helped us clarify the experimental objectives and limitations of our study, leading to substantial improvements in the overall quality of the revised manuscript.

Major points

Comment 1

Include (1) an L-cysteine-only control matched to CysNO, (2) at least one alternative NO donor (e.g., SNP or GSNO), and/or (3) an NO scavenger (e.g., cPTIO). These are needed to separate NO-dependent effects from donor- or cysteine-specific effects.

Response

Thank you very much for this important comment. We fully agree that including an L-cysteine-only control, an alternative NO donor, or an NO scavenger would further clarify the NO-specific effects. Several lines of evidence, however, indicate that the responses observed in this study primarily reflect nitric oxide–specific signaling.

In this study, CysNO was used at low concentrations (0.05–0.25 mM), far below those at which L-cysteine alone affects bacterial growth or metabolism (Koesnandar et al., 1990; Xie et al., 2013; Yan et al., 2022). In the previous study, CysNO treatment markedly induced hmp, which encodes the NO-detoxifying flavohemoglobin, confirming intracellular NO generation (Robinson & Brynildsen, 2013). Therefore, we focused on CysNO for this manuscript.

We have acknowledged the absence of these controls as a limitation and noted that our future studies on other microbial studies will include different NO donors and scavengers.

Koesnandar, A., Nishio, N., & Nagai, S. (1990). Stimulation by cysteine on growth of Clostridium thermoaceticum in minimal medium. Applied Microbiology and Biotechnology, 32(6), 711-714. https://doi.org/10.1007/BF00164746

Robinson, J. L., & Brynildsen, M. P. (2013). A Kinetic Platform to Determine the Fate of Nitric Oxide in Escherichia coli. PLOS Computational Biology, 9(5), e1003049. https://doi.org/10.1371/journal.pcbi.1003049

Xie, G.-J., Liu, B.-F., Xing, D.-F., Nan, J., Ding, J., & Ren, N.-Q. (2013). Photo-fermentative bacteria aggregation triggered by L-cysteine during hydrogen production. Biotechnology for Biofuels, 6(1), 64. https://doi.org/10.1186/1754-6834-6-64

Yan, X., Wang, X., Yang, Y., Wang, Z., Zhang, H., Li, Y., He, Q., Li, M., & Yang, S. (2022). Cysteine supplementation enhanced inhibitor tolerance of Zymomonas mobilis for economic lignocellulosic bioethanol production. Bioresource Technology, 349, 126878. https://doi.org/https://doi.org/10.1016/j.biortech.2022.126878

Comment 2

Unify the analysis across figures: define factors for ANOVA, report assumption checks (normality/homoscedasticity), multiple-comparison procedures, exact n per panel/metric, and whether effects are fixed or random. Ensure consistency between the text and figure legends: t-test figure 2) vs. Tukey (figure 4-6). Why did you use t-test in figure 2?

Response

Thank you for this helpful point. Figure 2 reports an OD600 time-course.

We revised the figure with a statistical analysis by two-way ANOVA with factors Time and CysNO concentration. Moreover, we kept the previous statistical results using Student’s t-tests, because we found a significant difference between the CysNO concentration. These results will help the potential reader to compare the effect on microbes from the CysNO supplementation. Please see below;

Figure 2. Effects of CysNO on B. vietnamiensis TVV75 growth with or without salt stress conditions. (A, B) Growth rate of B. vietnamiensis TVV75 at every 6 h interval in TSB (0.5% NaCl; control) or TSB + 2% NaCl supplemented with 0, 0.05, 0.1, and 0.25 mM of CysNO. Data are shown as mean ± SE (n = 4), and statistical analysis was performed by two-way ANOVA with Tukey test (p < 0.05). Student’s t-test was also performed to compare 0 mM with each CysNO concentration. Three independent experiments were carried out with similar results. Line 286-293

  Comment 3

Resolve inconsistencies in the control medium NaCl content across Methods and figure captions, and clarify whether “+2% NaCl” is total or added NaCl. This affects interpretation of all salt-response data.

Response

Thank you for pointing out the inconsistency in the description of NaCl content. We have clarified that the NaCl content “+2% NaCl” refers to the additional 2% NaCl to TSB instead of 2.5% NaCl. Accordingly, all sections have been revised to ensure consistency. Please see below;

3.3 CysNO reduced the cell death under NaCl stress

Given the observed growth inhibition under salt stress, we tested whether this was accompanied by reduced cell viability rather than just a growth reduction. We performed SYTO 9/propidium iodide (PI) live/dead staining and imaged with fluorescence microscopy. Viable cells emitted green fluorescence (SYTO9), whereas non-viable cells emitted red fluorescence (PI) [48]. Merged channel images were used to compare viability with or without salt stress supplemented with a series of CysNO concentrations. B. vietnamiensis TVV75 cultured in additional 2% NaCl showed a predominance of PI and less SYTO9 fluorescence compared with the absence of salt stress (Figure 3A). In contrast, CysNO-treatments largely reduced the PI fluorescence in both the presence and absence of salt stress conditions, indicating NO mitigates the salt-induced negative impacts on cell growth and viability.

Quantitative analysis of the SYTO9/PI fluorescence area ratio of these fluorescence signals supported the results of the microscopy images. TSB alone, with no CysNO-treated groups, had approximately 30% of PI-positive cells, indicating the presence of underlying membrane damage (Figure 3B). As the CysNO treatment concentration increased from 0.05 to 0.25 mM, the percentage of PI-positive cells decreased, falling below 10% at 0.05 mM, and remained low until 0.25 mM (Figure 3B). On the other hand, under additional 2% NaCl salt stress conditions, the PI-positive cells increased significantly to about 40%, indicating a significant increase in salt stress-induced cell death. However, the PI-positive percentage gradually decreased with the increase of CysNO concentrations, indicating that salt-induced cell death was notably alleviated by NO (Figure 3B). Line 254-276

Comment 4

Report activity units explicitly (e.g., U·mg⁻¹ protein - fig. 5B), how protein was quantified for each assay, and define clearly what is measured in intracellular vs. extracellular fractions (lysate vs. supernatant). In text we read " One unit (U) of SOD was defined as the amount causing 50% inhibition of NBT reduction. Activity was expressed as U/mL based on a 0.05 mL extract volume", but in figure 4C we can see - U mg protein. Is it correct?

Response

Thank you very much for your important observation. In the SOD assay, enzyme activity was measured from 0.05 mL of the cell lysate (intracellular fraction). Protein concentration in each enzyme extract was determined using the Bradford method, and the enzyme activity was normalized to total protein content. Accordingly, SOD activity is expressed as U mg⁻¹ protein in Figure 4C, which accurately represents the specific activity of the enzyme. The text in the Methods section (lines 199-200) has been revised to clarify this normalization and to explicitly indicate that the lysate was used for intracellular activity assays. Please see in below;

Superoxide dismutase (SOD, EC 1.15.1.1) activity was assessed by measuring the inhibition of NBT photoreduction [41]). A reaction mixture containing 50 μL of enzyme extract, 13 mM methionine, 75 μM nitroblue tetrazolium (NBT), 2 μM riboflavin, and 0.1 mM EDTA in 50 mM potassium phosphate buffer (pH 7.8) to a final volume of 1 mL was prepared. The mixture was incubated at 25 °C under 4000 lux light for 15 min. Absorbance was read at 560 nm using 200 μL of the reaction mixture. A sample without light or enzyme served as the blank; 50 mM potassium phosphate buffer (pH 7.8) was used as the positive control. One unit (U) of SOD was defined as the amount causing 50% inhibition of NBT reduction. Activity was expressed as U mg⁻¹ protein, normalized to the protein content determined by the Bradford assay. Line 181-190

Minor points

Comment 5

Where causality hinges on CysNO without a scavenger/alternate donor, soften to “support/suggest” rather than “demonstrate.”

Response

We have carefully revised the manuscript with the word “demonstrate” or “confirm” to “suggest” or “support”.

Comment 6

Unify 37°C or 37 °C (p. 3)

Response

We unified to “37°C ” in the revised MS.

Comment 7

Comments on the Quality of English Language

The manuscript is clearly structured but requires moderate editing for grammar, word choice, and stylistic consistency. A light professional edit by a fluent speaker is recommended.

  1. Use past tense for Methods/Results and present for general facts in Introduction/Discussion.
  2. Replace nonstandard phrases (e.g., “constitutive times” → “independent experiments”).
  3. Keep lists grammatically parallel (e.g., “measured enzyme activity, quantified gene expression, and assessed viability”).
  4. Use non-breaking space between value and unit (e.g., “25 °C,” “2% NaCl” without extra space before “%”); use a period for decimals.

Response

Thank you for pointing this out. We revised the following comments in the current MS.

Reviewer 2 Report

Comments and Suggestions for Authors

Dear Authors!

The sentences in the manuscript is not clear enough. My comments:

  1. You should make the sentence “CysNO supplementation also promotes the nitrate reduction pathway in vietnamiensis TVV75 under salt stress, suggesting NO-mediated nitrogen metabolism for microbial adaptation to salt stress.” more understandable (Abstract). For example, “CysNO supplementation also promotes the nitrate reduction pathway in B. vietnamiensis TVV75 under salt stress, suggesting the induction of NO-mediated nitrogen metabolism for microbial adaptation to salt stress.”
  2. I disagree with the sentence: “This membrane instability leads to electron leakage, which in turn induces oxidative stress [12,13].” (page 2). Probably you mean that the oxidative stress leads to oxidation of membrane lipids, membrane instability which in turn induces electron leakage from cells.
  3. You should decipher the abbreviations SOD, CAT (Abstracts), SNPs (page 2), TSA, PBS, BSA (page 3), TCA, TBC (page 4). Write uniformly HURiEN or HURIEN. Write about the products of lipA and ELFPP genes (Abstract).
  4. Are you writing about induction/reduction of enzymes` activity or their genes` expression in the sentence “Despite this stress-reducing and hydrolytic enzyme-inducing potential of NO, the stress-reducing effects of NO in microbes under salt stress remain poorly understood. Rephrase it to make more understandable.
  5. Rephrase the sentence: “The control was cultured in tryptic soy broth (TSB) (Difco, USA) with 0.05% NaCl, while salt stress was cultured in TSB and TSA with 2% NaCl.” (page 3). For example: “The control was cultured in tryptic soy broth (TSB) (Difco, USA) with 0.05% NaCl, while bacteria under salt stress were cultured in TSB and TSA with 2% NaCl.”
  6. You should add the name  Doerner to the sentence “Enzyme extraction was done following the methodology of Doerner et al. (2017) [37].” (page 3).
  7. I didn`t understand how “…CysNO may used as a proxy for hydrolytic efficiency of lipase activity  relevant to food-waste processing.” (page 9). You should check the meaning of the sentence, may be rephrase it.
  8. Probably, you should rephrase the sentence ”Consistently, the relative expression of lipA and ELFPP increased under salt stress with CysNO concentration” to “Consistently, under salt stress the relative expression of lipA and ELFPP increased with the elevation ofCysNO concentration” (page 10).
  9. Rephrase the sentence: “The increased H2O2  and MDA, along with increased dead cells observed in our experiments, can be interpreted as a result of this Fenton-weighted oxidative load” (page 11). For example, “The increased H2O2  and MDA, along with increased dead cells observed in our experiments, can be interpreted as a result of this Fenton-weighted oxidative load anddisruptions of the mitochondrial electron transport chains”.
  10. It should be noticed that Burkholderia vietnamiensisare highly abandoned in the lungs of patients with cystic fibrosis, so this bacteria may be dangerous in some cases.

The article is relevant for the field and presented in a well-structured manner. The references are 52% out of the last 5 years, but I think they are relevant. It does not include an excessive number of self-citations. The manuscript is scientifically sound and the experimental design is appropriate to test the hypothesis. The manuscript’s results are reproducible based on the details given in the methods section. The figures/tables/images/schemes are appropriate. They properly show the data, easy to interpret and understand. The data are interpreted appropriately and consistently throughout the manuscript. The statistical analysis or data acquired from specific databases are correct and sufficient. The conclusions are consistent with the evidence and arguments presented. I did not find any ethical violations.

Comments on the Quality of English Language

Dear Authors!

Some sentences need correction. They are unclear, as if some words are missing. I have mentioned them in comments above.

Author Response

Reviewer 2

We sincerely thank you for the careful reading and constructive suggestions, which have greatly improved the clarity and overall quality of our manuscript. All comments have been carefully considered, and the corresponding revisions have been incorporated into the revised version as detailed below.

Comment 1

The sentences in the manuscript is not clear enough. My comments:

You should make the sentence “CysNO supplementation also promotes the nitrate reduction pathway in vietnamiensis TVV75 under salt stress, suggesting NO-mediated nitrogen metabolism for microbial adaptation to salt stress.” more understandable (Abstract). For example, “CysNO supplementation also promotes the nitrate reduction pathway in B. vietnamiensis TVV75 under salt stress, suggesting the induction of NO-mediated nitrogen metabolism for microbial adaptation to salt stress.”

Response

Thank you for this helpful comment. We have revised this sentence accordingly. Please see below;

Abstract

Nitric oxide (NO), a reactive nitrogen species (RNS), plays a role in multiple biological functions and signal transduction. However, the mechanisms by which NO counteracts stress tolerance in microbes have been poorly explored. In addition, the decomposition of salty food waste poses a significant challenge for food-degrading microbes. Therefore, we investigated how S-nitrosocysteine (CysNO) affects the cellular salt stress response of Burkholderia vietnamiensis TVV75, a strain isolated from a commercial food waste composite. Under the additional 2% NaCl treatment, increased reactive oxygen species (ROS) inhibited bacterial cell growth and viability. In contrast, CysNO treatment alleviated the cellular ROS levels and growth inhibition by augmenting the superoxide dismutase (SOD) and catalase (CAT) activities. CysNO supplementation also promotes the nitrate reduction pathway in B. vietnamiensis TVV75 under salt stress, suggesting NO-mediated nitrogen metabolism for microbial adaptation to salt stress. Furthermore, CysNO restored the intracellular lipid-degrading lipase enzyme activities, which were compromised by salt stress alone. This restoration was accompanied by a concentration-dependent increase in the relative expression of the lipA (lipase A) and ELFPP (esterase lipase family protein) genes. These results suggest that external NO supplementation can regulate redox balance, nitrate reduction, and lipase activity to maintain microbial cell growth in high-salt environments, pinpointing a NO-dependent salt stress adaptation strategy for salt-sensitive microbes involved in food waste decomposition.  Line 14-33

Comment 2

I disagree with the sentence: “This membrane instability leads to electron leakage, which in turn induces oxidative stress [12,13].” (page 2). Probably you mean that the oxidative stress leads to oxidation of membrane lipids, membrane instability which in turn induces electron leakage from cells.

Response

We agree with this misinterpretation and have corrected the sentence accordingly. Please see below;

The salt content of Korean food waste is reported to be 2.19%, which is higher than in other countries [9], and is a negative factor for microbial-based food degradation. Salt stress induces osmotic imbalance, protein denaturation, and loss of membrane potential in microbial cells [10,11]. The membrane instability is due to the electron leakage caused by oxidation of the membrane [12,13]. This stress not only causes physical damage to the cell but also induces alterations in protein conformation and function, DNA impairment, and leads to decreased cell viability and enzyme activity [11,14], affecting the production of food-degrading enzymes [9,15].  Line 51-58

Comment 3

You should decipher the abbreviations SOD, CAT (Abstracts), SNPs (page 2), TSA, PBS, BSA (page 3), TCA, TBC (page 4). Write uniformly HURiEN or HURIEN. Write about the products of lipA and ELFPP genes (Abstract).

Response

Thank you for pointing this out. All abbreviations have been defined at their first appearance in the Abstract and the main text. The spelling of “HURIEN” has been unified throughout the manuscript. The Abstract now specifies that lipA and ELFPP encode lipolytic enzymes associated with lipid degradation. Please see below;

Abstract

Nitric oxide (NO), a reactive nitrogen species (RNS), plays a role in multiple biological functions and signal transduction. However, the mechanisms by which NO counteracts stress tolerance in microbes have been poorly explored. In addition, the decomposition of salty food waste poses a significant challenge for food-degrading microbes. Therefore, we investigated how S-nitrosocysteine (CysNO) affects the cellular salt stress response of Burkholderia vietnamiensis TVV75, a strain isolated from a commercial food waste composite. Under the additional 2% NaCl treatment, increased reactive oxygen species (ROS) inhibited bacterial cell growth and viability. In contrast, CysNO treatment alleviated the cellular ROS levels and growth inhibition by augmenting the superoxide dismutase (SOD) and catalase (CAT) activities. CysNO supplementation also promotes the nitrate reduction pathway in B. vietnamiensis TVV75 under salt stress, suggesting NO-mediated nitrogen metabolism for microbial adaptation to salt stress. Furthermore, CysNO restored the intracellular lipid-degrading lipase enzyme activities, which were compromised by salt stress alone. This restoration was accompanied by a concentration-dependent increase in the relative expression of the lipA (lipase A) and ELFPP (esterase lipase family protein) genes. These results suggest that external NO supplementation can regulate redox balance, nitrate reduction, and lipase activity to maintain microbial cell growth in high-salt environments, pinpointing a NO-dependent salt stress adaptation strategy for salt-sensitive microbes involved in food waste decomposition.  Line 14-33

We also measured transcriptional changes in the genes of osmC1/C2 (osmotically inducible protein C) and treZ (maltooligosyltrehalose trehalohydrolase), which are the proteins that help bacterial cells respond to internal pressure and damage under oxidative and osmotic stress, respectively. We found an increased activity of SOD and CAT with the supplementation of CysNO, particularly at 0.05 and 0.1 mM of CysNO (Figure 4C, D). However, at a relatively higher concentration (0.25 mM), their activities were decreased, which seems to contradict earlier findings on lower ROS production. The reduced ROS production may be directed by the upregulation of treZ at 0.25 mM of CysNO rather than the SOD and CAT activity (Figure 4G. Nevertheless, consistent with SOD and CAT activity, the relative expression of osmC1/osmC2 showed a significant upregulation by 0.1 mM of CysNO treatment (Figure 4E, F). Collectively, these results indicated that NO balances the redox switch in B. vietnamiensis TVV75 under salt stress, which mitigates the cellular damage and parallels a reduction in cell death. Line 295-307

Saline conditions heavily interrupt bacterial nitrogen metabolism, a crucial factor in maintaining a sustainable ecosystem [47,51]. In particular, nitrate reduction plays a major role in the nitrogen cycle, whereby nitrate is sequentially reduced to nitrite, N oxides (NO and N2O), and dinitrogen (N2) [52]. Therefore, to assess whether CysNO changes in nitrate reduction pathways (respiratory vs assimilatory) under salt stress, we quantified NR activity, extracellular NO2⁻, and transcriptional changes of narG, narH (membrane nitrate reductase), hmp (flavohemoglobin for NO detoxification), and nirB and nirD (assimilatory nitrite reductase) in B. vietnamiensis TVV75 (Figure 5). Results exhibited that CysNO treatments had a noteworthy modification in the reduction process with or without stress conditions. Line 321-330

Comment 4

Are you writing about induction/reduction of enzymes` activity or their genes` expression in the sentence “Despite this stress-reducing and hydrolytic enzyme-inducing potential of NO, the stress-reducing effects of NO in microbes under salt stress remain poorly understood. Rephrase it to make more understandable.

Response

Thank you for this helpful clarification. The sentence has been rephrased to specify that it refers to enzyme activity. Please see below;

Although NO has been reported to enhance microbial tolerance and stimulate hydrolytic enzyme activities, the stress-reducing effects of NO in microbes under salt stress remain poorly understood. Understanding how NO regulates the physiological and metabolic adaptations of microbes under salt stress is important as a new research axis in microbial-based food degradation. Line 81-85

Comment 5

Rephrase the sentence: “The control was cultured in tryptic soy broth (TSB) (Difco, USA) with 0.05% NaCl, while salt stress was cultured in TSB and TSA with 2% NaCl.” (page 3). For example: “The control was cultured in tryptic soy broth (TSB) (Difco, USA) with 0.05% NaCl, while bacteria under salt stress were cultured in TSB and TSA with 2% NaCl.”

Response

Thank you for this precise suggestion. The sentence has been corrected accordingly. Please see below;

2.2. NO-treated Bacterial Growth Assessment under Salt Stress

To assess the response of NO-mediated salt stress, the isolate was cultured in two conditions. The control was cultured in tryptic soy broth (TSB) (Difco, USA) with 0.5% NaCl, while bacteria under salt stress were cultured in TSB with 2% NaCl. This group simultaneously supplemented with 0, 0.05, 0.1, and 0.25 mM of CysNO. The additional 2% NaCl concentration was selected based on a preliminary test (see Supplementary Figure 1). TSB was cultured at 30°C, 150 rpm for 30 hours. OD600 value was measured every 6 hours. Tryptic soy agar (TSA) was incubated at 30 °C for 6 days, and colony diameters were measured using a digital caliper. Each condition was tested in triplicate, and the mean OD value and colony size were used to compare growth. Line 110-120

Comment 6

You should add the name  Doerner to the sentence “Enzyme extraction was done following the methodology of Doerner et al. (2017) [37].” (page 3).

Response

The citation has been corrected as suggested. Please see below;

Enzyme extraction was done following the methodology of Doerner et al. [37]. Briefly, after 30 h of bacterial incubation, cultures were centrifuged at 3000 rpm for 20 min at 4°C, and the supernatant was collected as the extracellular enzyme fraction. Line 135-137

Comment 7

I didn`t understand how “…CysNO may used as a proxy for hydrolytic efficiency of lipase activity relevant to food-waste processing.” (page 9). You should check the meaning of the sentence, may be rephrase it.

Response

Thank you for this valuable suggestion. The sentence has been revised for clearer meaning. Please see below;

3.6 CysNO application improved lipase activity under NaCl stress

With the observation of CysNO-induced NaCl tolerance in B. vietnamiensis TVV75, lipase activity was quantified under salt-stress conditions to verify whether CysNO supplementation could restore salt-inhibited lipid degradation and enhance food-waste degradation efficiency. We quantified lipase activity with p-nitrophenyl palmitate (pNPP) as a substrate, measured lipase-related gene expression, and performed tributyrin agar assays. Line 349-355

Comment 8

Probably, you should rephrase the sentence ”Consistently, the relative expression of lipA and ELFPP increased under salt stress with CysNO concentration” to “Consistently, under salt stress the relative expression of lipA and ELFPP increased with the elevation ofCysNO concentration” (page 10).

Response

The sentence has been revised accordingly. Please see below;

Intracellular activities were determined from cell lysates, and culture supernatants were analyzed for extracellular activities. Under salt stress, intracellular lipase activity increased 4-fold relative to the control condition, where CysNO further enhanced these intracellular activities in a concentration-dependent manner across the tested range (Figure 6A). Consistently, under salt-stress conditions, the relative expression of lipA (triacylglycerol lipase A) and ELFPP (esterase lipase family protein) increased with the elevation of CysNO concentrations (Figure 6B, C). Line 356-362

Comment 9

Rephrase the sentence: “The increased H2O2  and MDA, along with increased dead cells observed in our experiments, can be interpreted as a result of this Fenton-weighted oxidative load” (page 11). For example, “The increased H2O2  and MDA, along with increased dead cells observed in our experiments, can be interpreted as a result of this Fenton-weighted oxidative load and disruptions of the mitochondrial electron transport chains”.

Response

Thank you for this detailed recommendation. The sentence has been revised for accuracy and clarity. Please see below;

The increased H2O2 and MDA, along with increased dead cells observed in our experiments, can be interpreted as a result of this Fenton-weighted oxidative load and disruptions of the bacterial respiratory electron transport chains. On the other hand, CysNO significantly lowered H2O2 and MDA levels. These changes induced by NO treatment can be explained by the previous studies [32,39]. Line 414-419

Comment 10

It should be noticed that Burkholderia vietnamiensisare highly abandoned in the lungs of patients with cystic fibrosis, so this bacteria may be dangerous in some cases.

Response

Thank you for this important comment. We have addressed this concern by adding the following statement to the Discussion. Please see below;

Furthermore, the strain used in our study, B. vietnamiensis, belongs to the Burkholderia cepacia complex (BCC), a group comprising both environmental and opportunistic clinical species. Within this complex, B. vietnamiensis is considered to exhibit relatively low pathogenic prevalence, as infections in cystic fibrosis patients have been reported only occasionally [75]. Thus, caution is advised for potential use in large-scale or commercial applications. Aside from this biosafety consideration, this study is limited as only an in vitro-level experiment was conducted under controlled culture conditions, which does not fully reflect the complex environmental variables, such as temperature, oxygen concentration, substrate composition, and microbial interactions. Future studies should validate the applicability of the mechanisms proposed in this study by evaluating the effects of NO treatment in consortium-based pilot-scale systems that mimic the environment of a real household digester. In addition, to extend the understanding of the mechanisms of NO in salt-sensitive, food-waste-degrading microorganisms and to use them in practical food-waste processing, hmp or NR-deleted strains or the application of NO scavenging product will be a possible approach. Such future studies will further support the potential application of nitric oxide-mediated enhancement of microbial salt tolerance in real industrial conditions. Line 497-512

Comment 11

The article is relevant for the field and presented in a well-structured manner. The references are 52% out of the last 5 years, but I think they are relevant. It does not include an excessive number of self-citations. The manuscript is scientifically sound and the experimental design is appropriate to test the hypothesis. The manuscript’s results are reproducible based on the details given in the methods section. The figures/tables/images/schemes are appropriate. They properly show the data, easy to interpret and understand. The data are interpreted appropriately and consistently throughout the manuscript. The statistical analysis or data acquired from specific databases are correct and sufficient. The conclusions are consistent with the evidence and arguments presented. I did not find any ethical violations.

Response

We sincerely thank the reviewer for their positive comments regarding the manuscript's structure, scientific soundness, and appropriateness of the experimental design and conclusions.

Comment 12

Comments on the Quality of English Language

Dear Authors!

Some sentences need correction. They are unclear, as if some words are missing. I have mentioned them in comments above.

Response

Thank you for your critical feedback. We have followed and revised accordingly.

Reviewer 3 Report

Comments and Suggestions for Authors

Dear authors,

your manuscript, "S-nitrosocysteine ​​Modulates Nitrate-Mediated Redox Balance and Lipase Enzyme Activities in Food-Waste Degrading Burkholderia vietnamiensis TVV75 to Deter Salt Stress," demonstrates the effect of an exogenous NO on bacteria involved in the degradation of food waste derived from saline raw materials. Such research has both basic and applied significance, as it reveals mechanisms of salinity tolerance and can potentially provide recommendations for optimizing waste processing biotechnologies. You have reviewed your experiment critically and outlined the limits of the results.
The study is well-designed and meticulously executed. The conclusions are consistent with the results. Potential perspectives for further research are outlined.
This material can certainly be published.
I believe the following minor corrections could be made to improve the article:
1. Please indicate why you used CysNO as the source of NO, rather than other compounds such as SNP (sodium nitroprusside)? What are the advantages of CysNO?
2. What was the initial NO level in the cell culture before salinization and under the influence of NaCl?
3. Whenever gene names are mentioned for the first time, please, explain which proteins they encode.
4. Please explain the discrepancy between the increase in lipase activity and the decrease in lipA gene expression under salinization in the presence of 0.25 mM CysNO.

Overall, the results are well discussed, and the references are correct.
I hope you will be able to correct the manuscript.

Author Response

Reviewer 3

Dear authors,

your manuscript, "S-nitrosocysteine ​​Modulates Nitrate-Mediated Redox Balance and Lipase Enzyme Activities in Food-Waste Degrading Burkholderia vietnamiensis TVV75 to Deter Salt Stress," demonstrates the effect of an exogenous NO on bacteria involved in the degradation of food waste derived from saline raw materials. Such research has both basic and applied significance, as it reveals mechanisms of salinity tolerance and can potentially provide recommendations for optimizing waste processing biotechnologies. You have reviewed your experiment critically and outlined the limits of the results.
The study is well-designed and meticulously executed. The conclusions are consistent with the results. Potential perspectives for further research are outlined.
This material can certainly be published.
I believe the following minor corrections could be made to improve the article:

We sincerely thank you for the careful reading and constructive suggestions, which have greatly improved the clarity and overall quality of our manuscript. All comments have been carefully considered, and the corresponding revisions have been incorporated into the revised version as detailed below.

Comment 1

Please indicate why you used CysNO as the source of NO, rather than other compounds such as SNP (sodium nitroprusside)? What are the advantages of CysNO?

Response

Thank you very much for your insightful question. We selected CysNO because it is an endogenous S-nitrosothiol (SNO) that naturally occurs in biological systems and is therefore suitable for studying physiological NO-mediated interactions (Al-Sa'doni & Ferro, 2004).

Unlike sodium nitroprusside (SNP), which releases cyanide as a toxic by-product (Hirai et al., 2013), CysNO generates non-toxic reaction products, providing a safer and more physiologically consistent source of nitric oxide.

We added the advantages of CysNO in the revised MM. Please see below;

Although NO has been reported to enhance microbial tolerance and stimulate hydrolytic enzyme activities, the stress-reducing effects of NO in microbes under salt stress remain poorly understood. Understanding how NO regulates the physiological and metabolic adaptations of microbes under salt stress is important as a new research axis in microbial-based food degradation. In this study, we focused on analyzing the molecular mechanisms by which the application of exogenous S-nitrosocysteine (CysNO) can improve the salt tolerance of food-degrading bacteria. CysNO has been widely used in mammalian cells and plant studies, demonstrating its broad applicability as a biologically relevant NO donor with non-toxic characteristics compared to SNP. We isolated a single strain through screening from a commercial food waste decomposer preparation (HURIEN, Korea) used in household microbial food processors. Moreover, we performed a series of experiments to assess bacterial growth, cell viability, ROS production, nitrogen metabolism, and lipase activity under salt-stressed conditions with CysNO supplementation. Line 81-94

Al-Sa'doni, H., & Ferro, A. (2004). S-Nitrosothiols as Nitric Oxide-Donors: Chemistry, Biology and Possible Future Therapeutic Applications. Current medicinal chemistry, 11, 2679-2690. https://doi.org/10.2174/0929867043364397

Hirai, D. M., Copp, S. W., Ferguson, S. K., Holdsworth, C. T., Musch, T. I., & Poole, D. C. (2013). The NO donor sodium nitroprusside: evaluation of skeletal muscle vascular and metabolic dysfunction. Microvasc Res, 85, 104-111. https://doi.org/10.1016/j.mvr.2012.11.006

Comment 2

What was the initial NO level in the cell culture before salinization and under the influence of NaCl?

Response

We appreciate your insightful question. The initial NO level in the culture was not directly quantified; however, as shown in Figure 5E, the nitrite (NO₂⁻) concentration in the supernatant was measured as an indirect indicator of NO dynamics. Because nitric oxide in aqueous media is rapidly oxidized to nitrite, the low basal NO₂⁻ level observed in the control group (without NaCl and CysNO) suggests a minimal NO level of the cell culture before salinization.

Comment 3

Whenever gene names are mentioned for the first time, please, explain which proteins they encode.

Response

Thank you for your valuable suggestion. The full protein names encoded by the genes (narG, narH, hmp, nirB, nirD, lipA, and ELFPP) were already indicated in the corresponding figure captions and wherever needed. Please see below;

3.5 CysNO application altered the nitrogen metabolism to protect the bacterial cells under NaCl stress

Saline conditions heavily interrupt bacterial nitrogen metabolism, a crucial factor in maintaining a sustainable ecosystem [47,51]. In particular, nitrate reduction plays a major role in the nitrogen cycle, whereby nitrate is sequentially reduced to nitrite, N oxides (NO and N2O), and dinitrogen (N2) [52]. Therefore, to assess whether CysNO changes in nitrate reduction pathways (respiratory vs assimilatory) under salt stress, we quantified NR activity, extracellular NO2⁻, and transcriptional changes of narG, narH (membrane nitrate reductase), hmp (flavohemoglobin for NO detoxification), and nirB and nirD (assimilatory nitrite reductase) in B. vietnamiensis TVV75 (Figure 5). Results exhibited that CysNO treatments had a noteworthy modification in the reduction process with or without stress conditions. Line 318-330

Comment 4

Please explain the discrepancy between the increase in lipase activity and the decrease in lipA gene expression under salinization in the presence of 0.25 mM CysNO.

Thank you very much for your insightful comment. We addressed the discrepancy between the increase in total lipase activity and the decrease in lipA expression under 0.25 mM CysNO treatment.

As clarified in the revised Discussion, this difference likely results from the temporal gap between the rapid turnover of mRNA and the longer stability of enzyme proteins. Thus, even when lipA transcription declines, pre-existing lipases can sustain high total activity. In addition, B. vietnamiensis carries several lipolytic enzymes, including ELFPP and other putative esterases or phospholipases, which likely for the increment of lipase activity. This explanation has been added to the revised Discussion. Please see below;

This stabilization at the cellular level is manifested in the recovery of functional output-lipid hydrolysis-performance as shown in Figure 6. Intracellular lipase activity increased 4-fold under salt conditions and was further elevated by CysNO treatment, and lipA and ELFPP transcripts increased at 0.05 mM and 0.1 mM CysNO. However, at 0.25 mM CysNO treatment, lipA transcription levels decreased under salt stress, but total intracellular lipase activity increased. This discrepancy may reflect the inherent differences between mRNA and enzymes. The half-life of bacterial mRNA is extremely short, whereas the enzyme is relatively stable. In the study of Bacillus sp. LBN 2 cultures, while lipase activity peaked at 48 hours, a substantial amount of enzyme protein persisted until 72 hours [70]. Therefore, the total activity of pre-accumulated enzymes can be measured at a high level even if lipA transcription is temporarily low at a specific point in time. Moreover, B. vietnamiensis gene encodes multiple lipolytic enzymes, including ELFPP, that can likely contribute to the total lipase activity, rather than a single lipA product. On tributyrin agar, salt decreased colony size and absolute clear zone, but increased activity index (AI), which was further enhanced by CysNO. This suggests that salt stress may reduce the absolute amount of secreted enzyme through growth inhibition, but CysNO restored membrane stability and viability and optimized nitrogen/NO metabolism, resulting in higher efficiency per unit biomass. Similar to our results, Taskin et al. [34] reported that SNP treatment enhanced the lipase activity. Line 463-480

Comment 4

Overall, the results are well discussed, and the references are correct.
I hope you will be able to correct the manuscript.

We sincerely thank the reviewer for their positive comments on our manuscript. We have followed your suggestions and corrected.

Round 2

Reviewer 1 Report

Comments and Suggestions for Authors

The article can be accept in present form.